# A Twenty-Year Retrospective Analysis of Risk Assessment of Biomechanical Overload of the Upper Limbs in Multiple Occupational Settings: Comparison of Different Ergonomic Methods

**DOI:** 10.3390/bioengineering10050580

**Published:** 2023-05-11

**Authors:** Emma Sala, Lorenzo Cipriani, Andrea Bisioli, Emilio Paraggio, Cesare Tomasi, Pietro Apostoli, Giuseppe De Palma

**Affiliations:** 1Unit of Occupational Health, Occupational Hygiene, Toxicology and Prevention, University Hospital ASST Spedali Civili, 25123 Brescia, Italy; giuseppe.depalma@unibs.it; 2Unit of Occupational Health and Industrial Hygiene, Department of Medical Surgical Specialties, Radiological Sciences and Public Health, University of Brescia, 25123 Brescia, Italy; l.cipriani@unibs.it (L.C.); a.bisioli@unibs.it (A.B.); e.paraggio@unibs.it (E.P.); cesare.tomasi@unibs.it (C.T.); pietro.apostoli@unibs.it (P.A.)

**Keywords:** upper limb, biomechanical overload, risk assessment

## Abstract

Background: Several methods with which to assess the risk of biomechanical overload of the upper limb are described in the literature. Methods: We retrospectively analysed the results of the risk assessment of the biomechanical overload of the upper limb in multiple settings by comparing the application of the Washington State Standard, the threshold limit values (TLV) proposed by the American Conference of Governmental Industrial Hygienists (ACGIH), based on hand-activity levels (HAL) and normalised peak force (PF), the Occupational Repetitive Actions (OCRA) checklist, the Rapid Upper-Limb Assessment (RULA), and the Strain Index and Outil de Repérage et d’Evaluation des Gestes of INRS (Institut National de Recherche et de Sécurité). Results: Overall, 771 workstations were analysed for a total of 2509 risk assessments. The absence of risk demonstrated for the Washington CZCL, used as the screening method, was in good agreement with the other methods, with the sole exception of the OCRA CL, which showed at-risk conditions in a higher percentage of workstations. Differences in the assessment of the frequency of actions were observed among the methods, while their assessments of strength appeared to be more uniform. However, the greatest discrepancies were observed in the assessment of posture. Conclusions: The use of multiple assessment methods ensures a more adequate analysis of biomechanical risk, allowing researchers to investigate the factors and segments in which different methods show different specificities.

## 1. Introduction

Musculoskeletal disorders (MSDs) are major causes of morbidity among workers occupationally exposed to biomechanical overload. They comprise several signs and symptoms, such as pain, paraesthesia, fatigue, and limited range of motion, which can be related to work tasks. Workplace-related factors include physical, psychological, social, and biomechanical hazards. The main kinetic factors associated with MSDs include repetitive movements, excessive force, awkward postures, compression, and mechanical vibration. The correct evaluation of ergonomic hazard, musculoskeletal symptoms, and workplace exercise may help to reduce the occurrence of MSDs [1].

Several research studies have assessed the risk of the biomechanical overload of the musculoskeletal system based on a multimethodological comparative approach. Most of these studies analysed individual tasks in individual production sectors or contained descriptive analyses of the particular characteristics of risk-assessment methods [2,3,4,5,6,7,8,9,10,11,12,13,14,15,16,17,18,19,20]. There is a relative paucity of studies dealing with large case series for long observation periods [19,20,21].

Various tools are available for ergonomists and practitioners to assess the biomechanical overload of the musculoskeletal system. Among these, observational techniques have been used the most frequently, as they are relatively inexpensive, easy to use, and flexible, and they do not interfere with workers’ tasks while the use of direct measurement approaches, including motion capture/measurement, electronic goniometers, and push/pull force sensors, has minimally increased [22]. The present paper is an extension of our previous 10-year study [23]. The main objective of this study is therefore to present a 20-year experience of biomechanical risk assessments carried out through the simultaneous application of multiple observational methods described in the international body of research. An additional objective was to test the screening power of the Washington State Caution Zone Checklist (CZCL) [24] when used as a preliminary risk-assessment method. Finally, the degree of consistency among the applied methods is discussed, and useful operational information is provided with which to guide the choice of the most appropriate methods of analysis in different exposure contexts.

## 2. Materials and Methods

This is a retrospective 20-year study, including data collected through ergonomic surveys from 2003 to 2023, analysing 771 workstations in different manufacturing settings, through comparative application of multiple risk-assessment methods at different analytical levels, for an overall amount of 2509 assessments (Table 1).

Table 1 shows that most of the assessments were performed in factories producing various articles for children, including highchairs, car seats, strollers, baby bottles, and pacifiers.

The risk-assessment protocol included a multistep analysis process, as proposed by the guidelines of the Italian Society of Occupational Medicine [25] and the ISO 11228 technical standard [26], which started with a preliminary assessment conducted through verification of the risk items of the CZCL of the Washington Standard [24]. The tasks were then subjected to analysis by using the OCRA CL [27,28], HAL ACGIH [29], and RULA [30] as first-level methods, even in the absence of criticalities revealed at the preliminary assessment, since the objective of this analysis was also to confirm the screening capability of the Washington Standard’s CZCL. For tasks characterised by criticalities emerging using these first-level methods, further in-depth risk assessments were then conducted using higher-analytical-level methods (OREGE, Strain Index, and Washington State Standard’s Hazard Zone Checklist) [24,31,32,33].

We extended the evaluations already conducted in the period of 2003–2013 [23] to a larger case study, collecting data from twenty years of ergonomic surveys. Next, we compared the results obtained from the application of the different methods both as final risk indices and by individually highlighted risk factors.

The collected data were entered into a database and processed through “mathematical/algorithmic-informatic” classification methods proper to electronic Excel™ spreadsheets, based on “if-conditional” and control algorithms by summing and multiplying. The results obtained by the different methods were dichotomised as “at-risk” and “not-at-risk” and compared by two-by-two contingency tables with the Washington Standard’s CZCL method, using Fisher’s exact test with the statistical software GraphPad Prism ver. 9.5.1.

The cumulative frequency distributions of risk factors evaluated by the different methods were obtained by Excel™ software.

## 3. Results

Table 2 summarises the results of the assessments conducted by using the different methods. The preliminary assessment using the CZCL revealed an “at-risk situation” for 33% of the analysed tasks (N = 771). The OCRA CL was applied to a similar number of workstations (N = 765). In total, 38% of the tasks analysed with the OCRA CL were “at risk.” The at-risk situations accounted for 7% of the 464 workstations analysed by the HAL ACGIH method and for 10% of the 94 tasks evaluated by RULA.

The risk assessments carried out using second-level analysis methods, conducted on selected workplaces where a risk was highlighted by the assessment conducted using the methods described above, yielded the following results: the OREGE method, applied at 255 workstations, showed a situation from “not recommended” to “to be avoided” in 28% of cases; the Strain Index method, applied at only 165 workstations (this methodology is indeed not applicable in cases featuring a lack of technical actions in which force is applied), showed a picture ranging from uncertain to probably dangerous in 32% of the investigations. As shown in Table 2, most of the workstations analysed showed no significant risk for all the methods applied: in total, 63% of the workstations analysed with OCRA CL (rising to 82% when the workstations that were also found to have a borderline risk were included), 93% for the HAL-ACGIH, and 90% for the RULA. The second-level methods were only applied to the stations found to be at risk by the first-level methods and, therefore, the situations of no risk revealed by these methods were less frequent: 68% for the OREGE and 72% for the Strain Index. There was variability in the distribution of the intermediate-risk scores between the methods, which was also due to the variability of the intermediate-risk definition classes inherent in the structures of the methods themselves. The risk levels into which the synthetic indices were divided were in fact different and did not overlap with those of the methods applied: 5 for the OCRA CL, 3 for the HAL, 4 for the RULA, 3 for the OREGE, and 3 for the Strain Index. The percentages of the tasks found to be of high risk were superimposable for all the methods (0 to 2%), with the sole exception of the Strain Index method, which showed situations defined as “probably dangerous work” in 12% of the workstations. The Strain Index method, however, was only applied to the workstations that had already been found to be at risk for the hand–wrist anatomical area with the lower analytical level methods. In these cases, therefore the risk was confirmed by the method itself.

Table 3 shows that the results obtained by using the Washington CZCL and dichotomised as “risk items present” or as “no risk items” were not significantly different from the results obtained with the other first- and second-level methods, apart from those obtained using the OCRA CL method. As expected, the results of the Washington CZCL and those of the Washington HZ fully overlapped.

Next, we evaluated the cumulative frequency distributions of the frequency of actions and the strength in the workstations assessed by the HAL ACGIH, OCRA CL, and OREGE methods, which are summarised in Figure 1.

Figure 1 Cumulative frequency distributions of frequency of actions (upper graph) and strength scores (lower graph), ss evaluated by the OCRA CL, HAL ACGIH, and OREGE methods.

On the other hand, the lower graph in Figure 1 shows that the three evaluation methods were somewhat concordant in their evaluation of the risk factor “strength”, which is evaluated on a similarly graduated scale (0–10) in the three methods.

Figure 2 shows the cumulative frequency distribution of the recovery times, carried out using the OCRA CL. We can observe that in most of the workstations, the scores fell between 3 and 5 (55% of the total).

Table 4 shows the assessment of the postures carried out by using the different methods. When assessed by the OCRA CL, most of the workstations (75%) showed scores in the range of 0–2 (within a scale range of 0–47), with the mode represented by a score of 1 (47% of the assessments). When applying the OREGE method, the posture-assessment scores (scales 1–3) were lower overall, with 64% of the cases at level 1 (the lowest level in the method), 11% at 1.5, and 24% at 2. In the case of the assessments conducted with the Strain Index method, the posture scores (assessed exclusively for the wrist–hand area at five levels) reached the lowest level in 51% of the workstations and level 1.5 in 40% of them.

Table 5 shows the areas of the upper limb for which an overload was identified using the OCRA CL and OREGE methods. For the OCRA CL, the most negative evaluations were observed for the shoulder (in 51% of the workstations) and the hand (18%), followed by the wrist (15%); for the OREGE, the most negative evaluations were observed at the wrist level (in 70% of the analyses).

It should be noted that the tasks characterised by overloading at the level of the cervico-lumbar spine were only investigated with the Washington State Checklist (Caution Zone Checklist and Hazard Zone Checklist) and OREGE. Therefore, they were not comparable with the other methods, which did not include the assessment of spinal posture.

## 4. Discussion

The 2509 analyses carried out consisted mainly of a preliminary assessment, performed by using the Washington CZCL, followed by a more in-depth analysis, carried out, in most of the cases, by using the OCRA CL. The higher-analytical-level methods were applied at fewer workstations, as they were used for the purpose of analysing the critical situations highlighted in the preliminary analysis in greater depth. The choices of the methods applied were also dictated by the peculiarities and analytical limitations of the methods: the applicability of the Strain Index method, for example, is limited to the hand–wrist area and to force-engaging situations; the RULA method analyses static postures; and the HAL ACGIH is applied in single-task jobs and requires the measurement of manual activity levels and force.

Regarding the verification of the “screening capacity” of the Washington State Checklist items, we verified whether, for the tasks found to be at risk with the CZCL (33% of the total), the risk could be confirmed by using the other methods. On the other hand, the absence of risk when using the Washington CZCL was in good agreement with the higher-analytical-level methods, with the sole exception of the OCRA CL, which showed at-risk conditions in a higher percentage of workstations (38%). Thus, the Washington CZCL method showed a good negative predictive value for all the higher-level methods, apart from the OCRA CL. This lower concordance can be attributed to the nonuniform evaluation criteria of the observational methods used. For example, our analysis showed that the assessment of recovery time is crucial in the OCRA CL, but not in the other methods. In addition, the time required to maintain an incongruous shoulder posture, which is necessary for configuring risk with the OCRA CL, is very different from that in the other methods.

As in our previous experiences [23], we observed a greater correspondence between the preliminary risk assessment, the Washington State Hazard Zone Checklist, and the OREGE. This result can be explained by the fact that both the Washington Standard and the OREGE are used to investigate all the risk factors (strength, frequency, posture) in all the anatomical areas of the upper limbs (the wrist, elbow, hand, shoulder, and cervical spine). Some critical issues did not emerge with the other methods, which have more restricted scopes of application (as mentioned above, the Strain Index and ACGIH have topographic specificity for the hand–wrist area and for the risk factors of force and velocity of action). This may explain why, in some situations found to be at risk according to the preliminary assessment, the risk did not emerge in the next step.

Regarding the comparative analysis of the results obtained from the application of the different methods in our experience, we observed a good correspondence between the indices obtained from the application of all the methods in the extreme-risk ranges (absent or high), which was in agreement with the data in the literature [5,23]. Higher variability was found on the evaluations of the intermediate-risk ranges. There was variability in the distribution of the intermediate-risk scores between the methods, which was also due to the variability of the intermediate-risk definition classes inherent in the structures of the methods themselves. The risk levels into which the synthetic indices were divided were, in fact, different, and did not overlap across the different applied methods.

When comparing the assessment methods for the assessment of individual risk factors, awe observed different assessments of the frequency of action, which was of a higher degree with the ACGIH and OREGE than for the OCRA CL, although all the methods are based on a 0–10 scale. However, it must be emphasised that the criteria on which the measurement of repetitive movements and recovery times was based (along with the definition of movement/effort itself) were not homogeneous in the different methods under study:. The criteria were as follows: “Slow or very low motions with conspicuous long pauses, steady motion with frequent brief pauses, rapid steady motions with infrequent pauses “ for the HAL ACGIH; and ”short activity interrupted by long periods of breaks, slow and continuous movements with short pauses, continuous and regular with infrequent breaks etc. “ for the OREGE. For the OCRA CL, the definition of the frequency of action was based on ranges which were themselves based on the calculation of the number of actions per minute, and the definition of the recovery times was based on the calculation of the number of consecutive minutes of pause per hour. These intrinsic technical characteristics made it difficult to compare the criteria used in the methods for defining the frequency of action and recovery times. On the other hand, the assessments of strength appeared more uniform, based on 0–10 scales, with good agreement between the methods. These results confirmed that the literature data showed that an evaluation of the single components of the synthetic risk indices given by the methods is needed to evidence the specific critical aspects [9] and our previous experiences [23]. The greatest discrepancies were observed between the assessments of posture, since in the assessments of this risk factor, the various methods differed in terms of the duration of incongruous-posture maintenance necessary to configure risk, the level of association with other risk factors (e.g., strength), the range of motion, and the areas analysed.

We have already described, the differences in how individual risk factors were assessed between the methods in the first ten years of the ergonomic investigations under analysis.

In the analysis of posture using the Washington State Checklist, the overload of the shoulder region was configured if the posture was maintained for more than 2 h during a shift; for all the other areas, this duration was from 3 to 4 h (Hazard Zone Checklist). For the OCRA Checklist, risk was configured if the posture maintenance lasted for 10% of the cycle time, while for all the other districts, the posture was considered to present a risk according to the OCRA CL if it lasted for more than one-third of the cycle time. The RULA allows static-posture assessment, while the OREGE does not refer to the duration through defined criteria, but only highlights the need to increase the score by 1 if the posture is maintained for numerous minutes.

The Strain Index and the HAL ACGIH only allow the assessment of =the posture of the hand–wrist area [34].

The OREGE method assigns a mark of 3 (joint area to be avoided) only to incongruous shoulder posture; for the other areas, only notes 1 (comfort) and 2 (not recommended) are provided. The method, however, also allows the assessment of the cervical spine, unlike the OCRA CL, which only allows analyses that are focused on the upper limbs. The methods that allow the assessment of all the areas of the body (upper limbs, lower limbs, and spine) are, in our experience, the Washington State Checklist and the RULA method.

It is also crucial to consider the “weight” attributed to the maintenance of incongruous posture in the absence of other critical issues or in association with other risk factors [35]. For the Washington State Checklist, overhead shoulder posture is primarily considered to present a risk even in the absence of an association with repeated force, elbow posture is associated with repetitiveness, and wrist and hand posture present a risk if they are associated with force engagement and repetitiveness. For the spine and lower limbs, the element of association with incongruous spine posture is the absence of support or the possibility of varying the position held.

For the Strain Index method, incongruous posture in the hand–wrist area presents a risk only if it is associated with force engagement.

For the OCRA CL, unlike any of the other methods, incongruous posture in the various upper-limb areas presents a risk even in the absence of an association with force yes repetitiveness.

In relation to the association between strength and incongruous postures, the OCRA CL defines pinch grip as a risk factor independently of the exerted force. In our previous experience, in agreement to other research data [36], we demonstrated that muscle recruitment during pinch grip varies as a function of spontaneous force, and that in the evaluation of UL-WMSDs, the association between risk factors is essential to assess the actual extent of the overall risk: not only the position, but also the exerted force should be considered when assessing the pinch grip as a risk factor for the biomechanical overload of the upper limbs [37].

In the assessment of the risk of the biomechanical overload of the musculoskeletal system, the importance of assessing simultaneous exposure to several risk factors has long been known. The greater the simultaneous exposure to several biomechanical-overload risk factors, the greater the evidence of an association between work activity and MSDs [25,37,38].

Finally, it should be mentioned that the level of risk attributed to the degree of joint-angle deviation from neutral posture varies across the different methods. According to the OCRA CL, for example, the shoulder assumes an incongruous posture when the arm is raised over 80°; according to the Washington Checklist, this occurs when the hands are above the head. We believe that such inconsistencies will be resolved when observational ergonomic methods are paired with the simultaneous analysis of objective parameters by wearable sensor technologies.

The analysis of the 2509 risk assessments conducted over these two decades reconfirmed the analytical peculiarities, which are difficult to compare, of the various observational assessment methods already highlighted in our previous experiences [23]. In agreement with the literature data [2,3,4,5,6,7,8,9], the specificities of the different methods for each risk factor investigated and for the anatomical districts of the upper limbs that they are used analyse are fundamental to the choice of the most suitable method for the analysis of a task and the study of preventive measures.

This study also highlighted the importance of risk assessments conducted through multistep pathways, starting with a preliminary assessment using a validated method. Indeed, this study, in line with our previous experience, highlights that the Washington tool is a good screening method.

We believe that our contribution can help the ergonome to choose the correct method for biomechanical-overload-risk assessment, and that it highlights the importance of analysing each individual risk factor for each anatomic district of the upper limb.

Despite the limitations of this study, including the use of observational methods, which are by definition operator-dependent, as well as the different numbers and types of tools applied to the workstations analysed (indeed, the choice of method also depends on the characteristics of the tasks), in agreement with the literature data [1,2,3,4,5,6,7,8,9,10,11,12,13,14,15,16,17,18,19,20,21,22,23,35,38], the present analysis confirms the importance of the assessment of the risk of biomechanical overload of the upper limbs, based on the following basic principles: the correct choice of the most suitable assessment methods with which to study each specific task; a comparative analysis of the results obtained from the application of several methods; and, finally, the analysis of each of the risk factors and the levels of association between them to obtain a better, real, and complete estimation of risk.

Future developments of this research will be aimed at the progressive objectification of each individual risk factor by means of assessment tools, such as kinematic sensors, surface electromyography, load cells, opto-electronic systems, etc.

This will make it possible to translate the method of risk assessment by using a multi-methodological comparative approach to risk assessment, based on a comparison observational and objective methods.

## Figures and Tables

**Figure 1 bioengineering-10-00580-f001:**
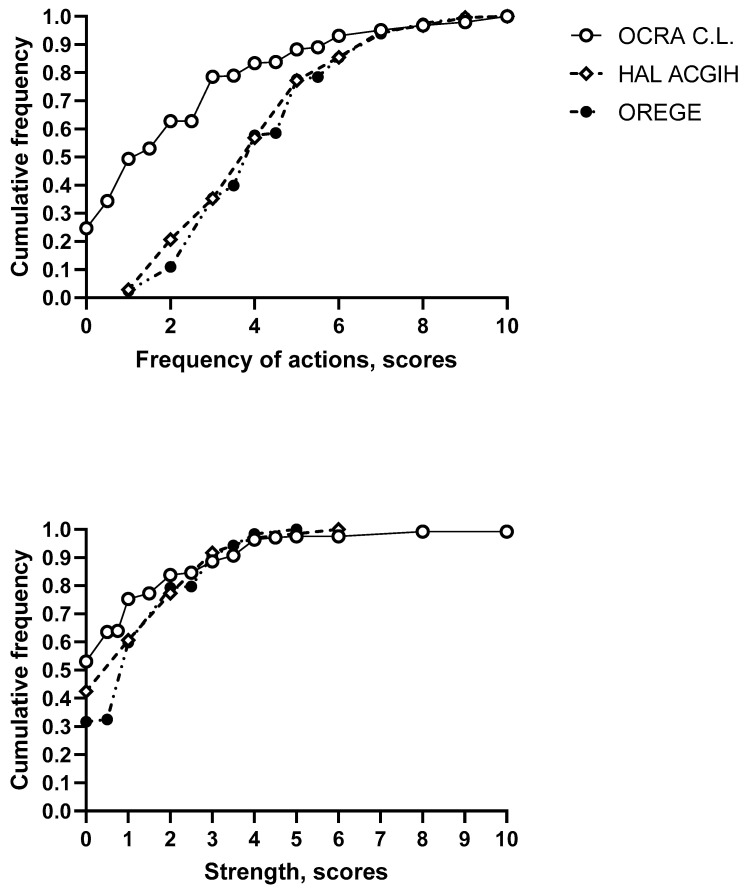
Upper graph shows that although the evaluations conducted by the HAL ACGIH and OREGE methods were structured on 0–10 analogy scales, as with the OCRA CL, they overlapped, but were different from those of the OCRA CL. In particular, the OCRA CL undervalued the frequency of actions compared to the other two. As can also be observed graphically, the assessments were mostly discordant in terms of the risk scores for intermediate frequencies, whereas the assessments were very similar, almost overlapping, in terms of the extreme scores, especially at higher values.

**Figure 2 bioengineering-10-00580-f002:**
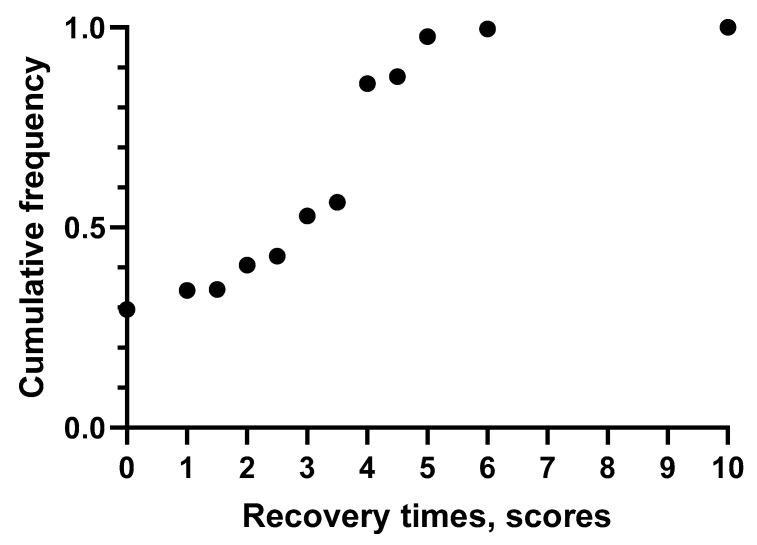
Evaluation of recovery times by OCRA CL checklist.

**Table 1 bioengineering-10-00580-t001:** Manufacturing settings used as objects of risk assessment.

Manufacturing Settings	Workstations	
	N	%
Childcare articles	357	46
Automotive	176	23
Alimentary	61	8
Textile	52	7
Metallurgical	37	5
Industrial electronics	24	3
Logistic	13	2
Metalmechanics	12	2
Pipe factory	11	1
School canteen	10	1
Wellness	9	1
Assembly of large household appliances	3	<1
Plastic manufactoring	3	<1
Mortuaries	1	<1
Household products	1	<1
Global logistics	1	<1
Total	771	100

**Table 2 bioengineering-10-00580-t002:** Distribution of results obtained by applying the different methods. Gray boxes indicate at-risk situations.

Methods (Workstations, N)	Results																					
WASHINGTON CZCL (771)	No-Risk Items										Risk Items Present											
	N					%					N						%					
	518					67					253						33					
OCRA CL (765)	Acceptable				Very light					Light		Medium						Intense				
	N	%			N			%		N	%	N			%			N				%
	478	63			114			19		66	9	63			8			14				2
HAL ACGIH (464)	1 (<Action Level)								2 (Action Level < X < TLV^®^)					3 (>TLV^®^)								
	N			%					N		%			N						%		
	430			93					26		6			8						2		
RULA (94)	1 (Action level 1)					2 (Action level 2)					3 (Action level 3)					4 (Action level 4)						
	N		%			N			%		N		%			N					%	
	85		90			9			10		0		0			0					0	
OREGE (255)	1 (Acceptable)								2 (Not Recommended)					3 (To Avoid)								
	N			%					N		%			N					%			
	183			72					72		28			0					0			
STRAIN INDEX (165)	1 (Probably Safe)								2 (Uncertain Evaluation)					3 (Probably Dangerous)								
	N			%					N		%			N					%			
	113			68					33		20			19					12			
WASHINGTON HZ (771)	No Risk										At risk											
	N						%				N					%						
	764						99				7					1						

**Table 3 bioengineering-10-00580-t003:** Distributions of the preliminary results obtained with the Washington CZCL method and of further assessments using first- and second-level methods.

Methods	No Risk Items	Risk Items Present	Total	*p* Fisher Exact Test
Washington CZCL vs. OCRA CL	707 vs. 478	58 vs. 287	765	<0.001
Washington CZCL vs. HAL ACGIH	433 vs. 430	31 vs. 34	464	0.797
Washington CZCL vs. OREGE	188 vs. 183	67 vs. 72	255	0.691
Washington CZCL vs. Strain Index	126 vs. 113	39 vs. 52	165	0.139
Washington CZCL vs. RULA	90 vs. 85	4 vs. 9	94	0.249
Washington CZCL vs. Washington HZ	5 vs. 5	2 vs. 2	7	>0.999

**Table 4 bioengineering-10-00580-t004:** Distributions of posture scores in investigated workstations using different methods.

Methods	Posture Score	Workstation	
		N	%
OCRA CL	0	29	4
	0.5	4	1
	1	353	47
	1.5	79	11
	2	93	12
	2.5	31	4
	3	33	4
	3.5	8	1
	4	51	7
	4.5	7	1
	5	17	2
	5.5	2	<1
	6	7	1
	7	7	1
	8	6	1
	9	5	1
	9.5	1	<1
	10	4	1
	11	4	1
	11.5	1	<1
	12	2	<1
	13.5	1	<1
	16	4	1
	18	1	<1
	19	1	<1
	TOTAL	751	100
OREGE	0	3	1
	1	161	64
	1.5	28	11
	2	60	24
	3	2	1
	4	1	<1
	TOTAL	252	100
STRAIN INDEX	1	83	51
	1.5	65	40
	2	14	9
	TOTAL	162	100

**Table 5 bioengineering-10-00580-t005:** Anatomical areas of the upper limb affected by biomechanical overload (1 = finger, 2 = wrist, 3 = elbow, 4 = shoulder) according to the evaluations by OCRA CL and OREGE methods.

Method	Overloaded Anatomical District	Workstation	
		N	%
OCRA CL	1	122	18
	2	103	15
	3	7	1
	4	355	51
	1-2	3	<1%
	1-2-3	3	<1%
	1-2-3-4	6	1
	1-2-4	8	1
	1-3	6	1
	1-3-4	10	1
	1-4	55	8
	2-3	4	1
	2-4	11	2
	2-3-4	3	<1%
	TOTAL	696	
OREGE	2	7	70
	4	3	30
	TOTAL	10	

## Data Availability

Not applicable.

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
