# Peer review of "A Twenty-Year Retrospective Analysis of Risk Assessment of Biomechanical Overload of the Upper Limbs in Multiple Occupational Settings: Comparison of Different Ergonomic Methods"

_bioengineering, 2023, doi:10.3390/bioengineering10050580_

Round 1

Reviewer 1 Report

It is a well-written manuscript on a topic that is not widely known in its current format. The approach seems to provide a novel vision of practical interest.

The abstract is brief, concise, and structured, showing the main findings with numerical data and objective conclusions.

The introduction is brief, clear synthetic and presents the hypothesis and objectives of the work in an adequate way.

The methodology is clear, well described and ensures reproducibility. The type of study, although it is not ideal, is still adequate, and clearly is the one that presents possibilities of realization.

The results are presented in a clear, concrete way with objectifiable numerical data. the tables are clear. It would be good to show measures of dispersion in the variables that allow it.

Figure 1 is interesting, but the chosen brands create an overlap that makes evaluation difficult. It is suggested to change the type of graph to one with lines of different strokes with points marked with smaller markers.

The discussion is clear, no significant biases are detected. The study's limitations and strengths should be incorporated as well as a discussion of future questions/remaining lines of research.

The bibliography is appropriate and well presented.

None

Author Response

Dear reviewer,

Best regards

Emma Sala

Reviewer 2 Report

Thank you for allowing me to review the paper "A twenty-year retrospective analysis of risk assessment of bio- 2 mechanical overload to the upper limb in multiple occupational settings: comparison among different ergonomic methods". 

The paper deals with an interesting topic within the journal's scope, but it lacks originality. The following concerns should be addressed before considering it for publication:

1. please explain better the new findings observed in the present paper and their implication during working activity;

2. comment further on the results and the difference with previously published papers;

3. define the future developments of this research and its translationality. 

Minor English revision

Author Response

Dear reviewer,

Best regards

Emma Sala
